

# Understanding the NaCl-dependent behavior of hydrogen production of a marine bacterium, *Vibrio tritonius*

Nurhidayu Al-saari[1,2], Eri Amada[1], Yuta Matsumura[1], Mami Tanaka[1], Sayaka Mino[1] and Tomoo Sawabe[1]

[1] Laboratory of Microbiology, Faculty of Fisheries Sciences, Hokkaido University, Hakodate, Japan
[2] International Institute for Halal Research and Training (INHART), International Islamic University Malaysia (IIUM), Kuala Lumpur, Malaysia

## ABSTRACT

Biohydrogen is one of the most suitable clean energy sources for sustaining a fossil fuel independent society. The use of both land and ocean bioresources as feedstocks show great potential in maximizing biohydrogen production, but sodium ion is one of the main obstacles in efficient bacterial biohydrogen production. *Vibrio tritonius* strain AM2 can perform efficient hydrogen production with a molar yield of 1.7 mol $H_2$/mol mannitol, which corresponds to 85% theoretical molar yield of $H_2$ production, under saline conditions. With a view to maximizing the hydrogen production using marine biomass, it is important to accumulate knowledge on the effects of salts on the hydrogen production kinetics. Here, we show the kinetics in batch hydrogen production of *V. tritonius* strain AM2 to investigate the response to various NaCl concentrations. The modified Han–Levenspiel model reveals that salt inhibition in hydrogen production using *V. tritonius* starts precisely at the point where 10.2 g/L of NaCl is added, and is critically inhibited at 46 g/L. NaCl concentration greatly affects the substrate consumption which in turn affects both growth and hydrogen production. The NaCl-dependent behavior of fermentative hydrogen production of *V. tritonius* compared to that of *Escherichia coli* JCM 1649 reveals the marine-adapted fermentative hydrogen production system in *V. tritonius*. *V. tritonius* AM2 is capable of producing hydrogen from seaweed carbohydrate under a wide range of NaCl concentrations (5 to 46 g/L). The optimal salt concentration producing the highest levels of hydrogen, optimal substrate consumption and highest molar hydrogen yield is at 10 g/L NaCl (1.0% (w/v)).

## INTRODUCTION

Hydrogen gas ($H_2$) has long been seen as an ideal alternative energy source due to its high energy content of 122 kJ/g and the fact it only produces water during combustion (*Atif et al., 2005*; *Chandrasekhar, Lee & Lee, 2015*; *Hao et al., 2006*; *Zhao et al., 2001*). Among many potential processes, biological $H_2$ (bio$H_2$) production via dark fermentation offers the best option that is both renewable and potentially viable for environmental and economic sustainability, allowing us to reduce our dependency on fossil fuels (*Akinbomi, Wikandari*

Corresponding author
Tomoo Sawabe,
sawabe@fish.hokudai.ac.jp

& Taherzadeh, 2015; Hallenbeck & Ghosh, 2009; Meinita et al., 2013; Myung et al., 2014). The third generation biofuels using marine algae as feedstock provide a great opportunity in maximizing biohydrogen production efficiency and could alleviate the ''food vs. fuel'' debate that arose from the utilization of terrestrial biomasses in the first and second generation biofuels (Alam, Mobin & Chowdhury, 2015; Goh & Lee, 2010; Singh, Nigam & Murphy, 2011). In comparison to land materials, marine biomass is less affected by climate change, less limited by land availability, has higher potential biomass yields per unit area and offers less competition to food supply (Milledge & Harvey, 2016; Milledge & Harvey, 2018). Marine macroalgae, in particular, lacks lignin, contains lipids, and a significant amount of sugars (at least 50% of its total dried weight); suitable attributes for bioH$_2$ production using biomass feedstock (Enquist-Newman et al., 2013; Wargacki et al., 2012; Takeda et al., 2011). The application of macroalgae in bioH$_2$ production, however, is potentially offset by the fact that the sugars in macroalgae, such as mannitol, are seasonal (Adams et al., 2011; Suutari et al., 2015; Yanagisawa, Kawai & Murata, 2013). Additionally, the algae are also rich in salt content with relatively more sodium chloride than terrestrial biomasses (Milledge & Harvey, 2016; Rupérez, 2002). High salt concentration has been reported as adversely affecting H$_2$ productivity in some bacterial species (Alshiyab et al., 2008). Continuous exploration and development of microorganisms that are capable of converting these special macroalgae carbohydrates to biohydrogen, preferably under saline conditions, would resolve these challenges and further enhance productivity.

To date, the majority of studies on fermentative H$_2$ production have been on strict anaerobes of *Clostridium* (Ferchichi et al., 2005; Jayasinghearachchi et al., 2010; Jo, Lee & Park, 2008b; Junghare, Subudhi & Lal, 2012), and facultative anaerobes such as *Escherichia coli* (Maeda, Sanchez-Torres & Wood, 2012; Trchounian & Trchounian, 2014; Trchounian et al., 2015) and *Enterobacter aerogenes* (Jo et al., 2008a; Zhao et al., 2009; Zhao et al., 2001). These terrestrial bacteria were not suitable biocatalysts in fermentative H$_2$ production under saline conditions due to sensitivity to salts in H$_2$ production (Alshiyab et al., 2008). In a screening of effective H$_2$ producers using seaweed carbohydrates, a facultative anaerobic marine bacterium, *V. tritonius* strain AM2, isolated from the gut of sea hare (*Aplysia kurodai*) shows higher H$_2$ production than those of enterobacteria (Sawabe et al., 2013; Matsumura et al., 2014). Production of gas during the fermentation of carbohydrates is an atypical property in the genus *Vibrio* with only *V. aerogenes*, *V. aphrogenes*, *V. furnissii*, *V. gazogenes*, *V. mangrovii*, *V. porteresiae*, *V. ruber*, *V. rhizosphaerae*, and *V. tritonius* among those sharing the attribute out of 140 *Vibrio* species. However, a series of experiments by Matsumura et al. (Matsumura et al., 2014; Matsumura et al., 2015) revealed unexpectedly high H$_2$ productivity by *V. tritonius* cultured in glucose- and mannitol-supplemented media under saline conditions (2.25% (w/v) NaCl). Surveys of the complete genome of *V. tritonius* strain AM2 shows a single 24-kb gene cluster containing 21 genes that is responsible for the formation of a formate hydrogen lyase (FHL) complex on the large chromosome. This vibrio FHL complex is structurally similar to the *hyf* gene cluster (*hyfABCDEFGHIJ* genes) found in *E. coli* K-12 with the amino acid identity of each homolog ranging from 45 to 68% (Matsumura et al., 2015). On the downstream of this vibrio *hyf* gene cluster, the bacterium also carries the genes responsible for the formate dehydrogenase (FDH-H), *fhlA*-type

transcriptional activator, and hydrogenase maturation proteins (*hyp*) which together form a "super-gene-set" of the FHL-Hyp gene cluster. The *V. tritonius* hydrogenase is also classified as a [NiFe]-hydrogenase because of typical motifs coordinating the [NiFe] center at the active site of its *hyfG*, the large subunit of FHL complex (*Matsumura et al., 2015*). The fermentation product profiling and the genome surveys suggest $H_2$ production from glucose and mannitol is via the FHL pathway (*Matsumura et al., 2014*; *Matsumura et al., 2015*).

*V. tritonius* AM2 is a slight halophile that is capable of growing in 0.5 to 8.0% (w/v) NaCl. Optimum growth of the bacterium was observed in 3.0% NaCl (w/v) (*Sawabe et al., 2013*). Thus, the marine bacterium potentially possesses different features relating to $H_2$ production compared to those observed in terrestrial bacteria. Application of seawater for bacterial fermentative production is still being considered as a cost-effective technology (*Adessi et al., 2016*; *Maeda, Sanchez-Torres & Wood, 2012*). The idea of using marine vibrio species could be an ideal solution in intensifying further progress being made for third generation biofuel production using seaweed as feedstock. *Vibrio* species can also be used in bio$H_2$ production when domestic and food industries residual wastes containing high salt concentrations are used as feedstock (*Cabrol et al., 2017*; *Oh et al., 2003*; *Pierra et al., 2014*). Previously, *V. tritonius* was reported as successfully producing $H_2$ from powdered seaweed using a jar fermenter under pre-optimized culture conditions (*Matsumura et al., 2014*). These results helped the idea that the use of seaweed for fermentation can be a cost effective and sustainable alternative because the fact seaweed for feedstock can be prepared using simple pre-treatment such as pulverization. The halophilic and/or halotolerant natures of *Vibrio* species can provide huge advantages in $H_2$ production and in reducing the use of freshwater for feedstock pre-treatment to remove salt (*Shi et al., 2013*).

Previous research on fermentative hydrogen production by *V. tritonius* concentrated on the effects of different pH levels, temperatures and substrates on higher hydrogen yields (*Matsumura et al., 2014*). The effects of salt concentration on hydrogen production by strain AM2, however, are not clearly defined and warrants further investigation. Herein, we investigate the effects of different NaCl concentrations on the formate-hydrogen conversion of *V. tritonius* AM2. We used a modified-Gompertz model to elucidate the kinetics of cell growth, hydrogen production and substrate consumption during the fermentation process in an attempt to improve hydrogen production and understand the process as NaCl levels increased. A modified Han–Levenspiel model simulates the relationship of hydrogen production rates in response to increasing NaCl concentration. We also compared the formate metabolism of *E. coli* strain JCM 1649, which possesses a similar FHL system in the fermentative $H_2$ production to those of *V. tritonius*. We cultured the *E. coli* strain JCM 1649 in media supplemented with different NaCl levels to those of the marine vibrio. This study is expected to help further the understanding of the physiological impact of NaCl on formate-hydrogen bioconversion via the FHL pathway.

## MATERIAL AND METHODS

### Bacterial strain and pre-cultivation

Pure culture of *V. tritonius* strain AM2 (JCM 16456$^T$) was maintained on a marine agar slant at 15 °C and sub-cultured 3 times a month. The cells were then pre-cultured aerobically in marine broth to reach a cell density of 0.8 to $1.0 \pm 0.1$ OD$_{620}$ measured by a microplate reader (Infinite F200, Tecan, Switzerland), which corresponds to 1.4 to $1.7 \pm 0.1$ OD$_{620}$ using a UV/Visible spectrophotometer (Ultrospec 2000; Pharmacia Biotech, Piscataway, NJ, USA), with shaking at 130 rpm (MMS-48GR; EYELA, Tokyo, Japan) at 30 °C. The pre-culture medium was a marine broth composed of 0.5% (w/v) polypeptone, 0.1% (w/v) yeast extract, and 75% (v/v) artificial seawater (ASW) composed of 0.7 g/L KCl, 5.3 g/L MgSO$_4$ · 7H$_2$O, 10.8 g/L MgCl$_2$ · 6H$_2$O, 1.3 g/L CaSO$_4$ · 2H$_2$O, and 30 g/L NaCl at pH 7.5.

In a separate experiment, *Escherichia coli* strain JCM 1649 was used as a reference. Similar to the experiment using *V. tritonius* AM2, the pure culture *E. coli* was stored on LB agar slant (1% (w/v) tryptone, 0.5% (w/v) yeast extract, 0.5% (w/v) NaCl, 1.5% (w/v) agar) at 15 °C and sub-cultured three times a month. Pre-culture was prepared using LB broth following a similar method to that used for *V. tritonius*.

### Experimental designs

Batch culture was performed using *V. tritonius* strain AM2 inoculated in a basal marine media made of 0.5% (w/v) polypeptone, 0.1% (w/v) yeast extract, 100 mM MES (Dojindo, Kumamoto, Japan), and 75% (v/v) basal salt solution composed of 0.7 g/L KCl, 5.3 g/L MgSO$_4$ · 7H$_2$O, 10.8 g/L MgCl$_2$ · 6H$_2$O, and 1.3 g/L CaSO$_4$ · 2H$_2$O. To study the effects of Na$^+$ concentration, the amount of NaCl was adjusted accordingly to constitute 0.5%, 1.0%, 2.25%, 3.0% and 5.0% (w/v) of the culture media. One mL of the precultured inoculum was then added to 95 mL marine broth containing 5% (w/v) mannitol in a glass bottle reactor. Batch culture was performed at 37 °C and pH 6.0 with gentle agitation using a magnetic stirrer (speed at 4.5, RO 10 power IKAMAG; IKA, Staufen, Germany). The pH was maintained using a pH controller (DT-1023P; ABLE, Tokyo, Japan) attached to an autoclavable electrode (FermProbe pH electrodes; Broadly-James Corp., Irvine, CA, USA) with the addition of 5 N NaOH. The experiment was repeated in triplicate and samplings of culture and gas were done at 12 h-intervals.

In batch culture experiments for *E. coli* strain JCM 1649, the bacteria was cultured on three types of media namely LB broth (1% (w/v) tryptone, 0.5% (w/v) yeast extract, 0.5% (w/v) NaCl), LB broth prepared using 60% (w/v) artificial seawater (with total NaCl equal to 2.30% (w/v) and denoted as LB broth with 60% ASW) and marine broth (containing 2.25% (w/v) NaCl). 100 mM MES (Dojindo, Kumamoto, Japan) was supplemented in all media. The strain JCM 1649 was precultured under the same conditions as the strain AM2 culture, which gave a cell density of 1.3 OD$_{620}$ after 24 h incubation. In these batch culture experiments, broth containing 5% (w/v) glucose was used for strain JCM 1649, due to its unsuitability in utilizing mannitol. Other culture conditions were set the same as those for the strain AM2. For both experiments using *V. tritonius* strain AM2 and *E. coli* strain JCM 1649, neither external reducing agents nor sparging with inert gas is required because

both bacteria are capable of self-installing and maintaining anaerobiosis for hydrogen production using sugars as carbohydrate (*Hassan & Morsy, 2015*; *Matsumura et al., 2015*).

## Assays

The biogas produced was collected in an aluminum-bag (GL Sciences Inc, Tokyo, Japan) and measured using the water displacement method. Hydrogen content in the biogas was determined using gas chromatography (GC-2014; Shimadzu, Kyoto, Japan) equipped with thermal conductivity detector (TCD) and ShinCarbon ST packed column (Shinwa Chemical Industries Ltd., Kyoto, Japan) with Argon (Ar) as the carrier gas. The operational temperatures of the injection port, oven and detector were 60 °C, 40 °C and 60 °C respectively. Extracellular ethanol was measured using the same gas chromatography equipment together with a flame ionization detector (FID) and HP-Blood Alcohol Capillary column (Agilent, Santa Clara, CA, USA). Helium (He) was used as the carrier gas. The temperatures of the column, oven and detector were 60 °C, 40 °C and 60 °C, respectively.

Organic acids produced, such as formate, acetate, lactate and succinate were measured using high performance liquid chromatography (HPLC) with a conductivity detector and tandem connected to two Shim-pack SCR-102H columns (Shimadzu, Kyoto, Japan). 5 mmol/L *p*-toluenesulfonic acid was used as the mobile phase with the flow rate at 0.8 mL/min. Mannitol concentration was measured by HPLC with a reflective index detector (RID) and TSKgel Amide-80 column (TOSOH, Tokyo, Japan). 75% acetonitrile was used as the mobile phase with the flow rate at 1.0 mL/min. The optical density (OD) of cell growth was measured using a microplate reader (Infinite F200; Tecan, Männedorf, Switzerland) at a wavelength of 620 nm.

## Data analyses

Molar yields of the fermentative products including hydrogen were analyzed using one-way analysis of variance (ANOVA) with Duncan's multiple range tests. All statistical analyses were performed using the Statistical Analysis System (SAS) University Edition. Probabilities of less than 5% ($P < 0.05$) were considered statistically significant.

The progress of cumulative hydrogen production, $H$ (mL), in batch tests can also be described using the following modified Gompertz model:

$$f(X) = \alpha.Xmax.\exp\left(-\exp[\frac{\alpha.\mu max.e}{\alpha.Xmax}(\lambda - t) + 1]\right). \tag{1}$$

Replacing $\alpha.Xmax$ with $P$ as hydrogen production potential (mL), and $\alpha.\mu max$ by $R$ m, as maximum hydrogen production rate (mL h$^{-1}$), Eq. (1) can be expressed by Eq. (2):

$$H = P.\exp\left(-\exp[\frac{Rm.e}{P}(\lambda - t) + 1]\right). \tag{2}$$

Similarly, the substrate consumption could also be expressed using the modified Gompertz model (*Mu, Yu & Wang, 2007*):

$$S_0 - S = S_{max}.\exp\left(-\exp[\frac{Rms.e}{S_{max}}(\lambda - t) + 1]\right). \tag{3}$$

With $S_0 - S$ as the amount of substrate consumed in g/L; $S_{max}$ as the maximum substrate consumed and $Rms$ as the maximum rate of substrate consumed (in g/L/h).

$\lambda$ is the lag time (h) and $e$ is equal to 2.718. All parameters in Eq. (2) were non-linearly estimated by converting the sum square of errors (SSE) between experimental data and estimation from the models to a minimum value. This estimation was carried out using the 'Solver' function (Euler integration technique) in Microsoft Excel 2010 (*Kemmer & Keller, 2010*). The significance of the estimated parameters was tested using analysis of variance (ANOVA) as mentioned above (*Chen et al., 2006*).

Hydrogen production rate, $r$ (mL h$^{-1}$) of AM2 grown at various salt levels was calculated following *Wang & Wan (2008)* in which:

$$r = \frac{P}{\lambda + \left(\frac{P}{Rm}\right)}. \qquad (4)$$

The hydrogen production rate was also denoted as HPR.

Response of the hydrogen production rate to different NaCl concentrations was calculated using a modified Han–Levenspiel model as described in *Van Niel, Claassen & Stams (2003)* and *Zheng & Yu (2005)* by taking on the logarithms;

$$ln(r) = nln\left(1 - \frac{C}{Ccrit}\right) + ln(r\max) \qquad (5)$$

where $r$ is the hydrogen production rate (mL/h); $r$ max is the maximum hydrogen production rate; $C$ is the added Na$^+$ concentration (g/L); $C$ crit is the critical Na$^+$ concentration at which H$_2$ production ceases (g/L) and $n$ is the degree of inhibition.

In order to describe the microbial growth in a batch culture, the Gompertz equation was again modified following *Zwietering et al. (1990)* and *Begot et al. (1996)*:

$$ln\left(\frac{N}{N_0}\right) = A. \exp\left(-\exp\left[\frac{\mu.e}{A}(\lambda - t) + 1\right]\right) \qquad (6)$$

where $A$ is the logarithmic increase of bacterial population; $\lambda$ is the lag time; $\mu$ is the maximal growth rate; and $t$ is the time.

## RESULTS

### Cumulative H$_2$ production, its kinetics and the growth of *Vibrio tritonius* at various salt levels

The cumulative hydrogen production at different NaCl concentrations over time was fitted with Eq. (2) and is shown in Fig. 1. The experiment confirmed the significant effects of various levels of NaCl on cumulative H$_2$ production using AM2. Substrate fermentation took place for 72 h and at the end of the process, the culture in 0.5 and 1.0% (w/v) NaCl produced the highest amounts of H$_2$, i.e., 7,831 $\pm$ 295 and 8,561 $\pm$ 602 mL/L reactor volume on average, respectively. On the other hand, cells grown in media supplemented with 2.25% (w/v) NaCl produced slightly less hydrogen at 6,693 $\pm$ 443 mL/L reactor volume followed by 3.0% and 5.0% (w/v) NaCl at 1,808 $\pm$ 79 and 3 $\pm$ 2 mL/L reactor volume, respectively. Table 1 shows the $P$ as hydrogen production potential (mL), $R$ m, as maximum hydrogen production rate (mL h$^{-1}$) and $\lambda$ (h) of the batch system in response to various salt levels. The results imply the strain AM2 is capable of hydrogen production under a wide range of NaCl concentrations, i.e., 0.5 to 3.0% (w/v). However, the amount of

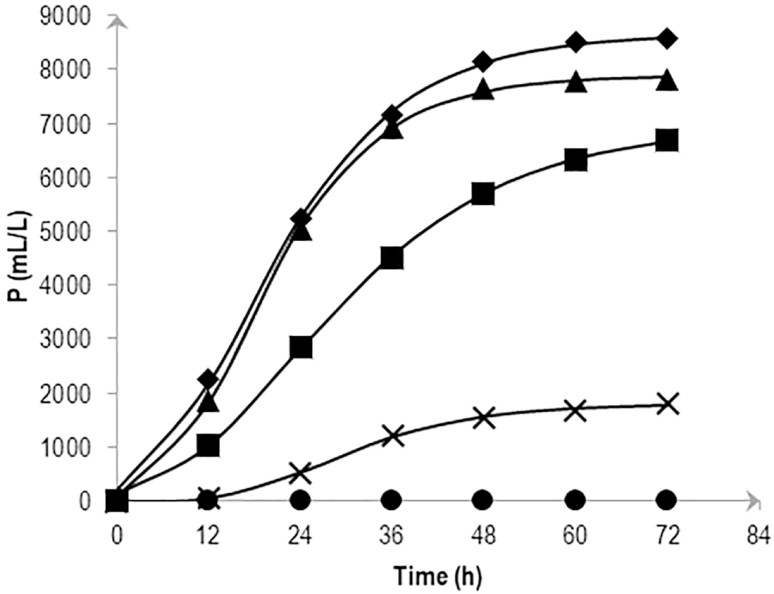

**Figure 1** **Effects of NaCl on the cumulative hydrogen production ($P$) of *Vibrio tritonius*.** The fermentation media containing 0.5% (♦); 1.0% (▲); 2.25% (■); 3.0% (X) and 5.0% (●) (w/v) NaCl were used. Symbols represent the experimental values and the curves were fitted with a modified Gompertz model Eq. (2) and drawn in full lines.

hydrogen produced dropped as NaCl concentrations increased. Maximum $H_2$ production potential, $P$, was observed when 0.5 and 1.0% (w/v) NaCl was supplemented in the media followed by the addition of 2.25% (w/v) NaCl and showed further decreases at higher NaCl concentrations in the media.

The efficiency of fermentative biohydrogen is strongly connected to the presence of active cells (*Mu, Wang & Yu, 2006*). Different NaCl concentrations greatly affected the growth of AM2. Cell growth significantly decreased beyond 2.25% (w/v) NaCl indicating the presence of some degree of inhibition with the addition of 3.0% and 5.0% (w/v) NaCl in the media. Statistical analysis of the growth at the end of fermentation (72 h) for 0.5 to 2.25% (w/v) NaCl supplemented media reveals that they were not significantly different ($P < 0.05$) to one another. We also calculate the maximum number of organisms, $X$ max, the maximum growth rate, $\mu_{max}$ [$\ln(N/N_0)$ h$^{-1}$] and the lag time, $\lambda$ (h) of the batch system in response to various salt levels. The maximum specific growth rate, $\mu_{max}$, was insignificant at 0.5 to 2.25% (w/v) with average $\mu_{max} = 0.8$ $\ln(N/N_0)$/h but inhibited at 3.0 and 5.0% (w/v) with an average of $\mu_{max} = 0.5$ $\ln(N/N_0)$/h. *V. tritonius* showed a classical growth trend under different salt concentrations and the average lag time, $\lambda$ was at 2.3 h. The cells enter a stationary growth phase after between approximately 14 to 23 h of fermentation and then plateaued until the end of the fermentation period. Table 2 summarizes the kinetics of bacterial growth at different NaCl concentrations as calculated using Eq. (6). The kinetics of bacterial growth and cumulative $H_2$ produced reveals that hydrogen production started during the exponential phase in which cell growth and $H_2$ formation may occur simultaneously. Later, hydrogen evolution continues during the

Al-saari et al. (2019), *PeerJ*, DOI 10.7717/peerj.6769

**Table 1** **Kinetic parameters of cumulative hydrogen production and mannitol consumption by *Vibrio tritonius* under batch fermentation at various salt levels calculated from a modified Gompertz model.**

| NaCl concentration (%)(w/v) | Cumulative hydrogen production | | | | Mannitol consumption | | |
|---|---|---|---|---|---|---|---|
| | Maximum H$_2$ production potential: P (mL/L reactor volume) | Maximum H$_2$ production rate: Rm (mL/h) | Lag time: λ (h) | Hydrogen production rate (HPR): r (mL/h) | Maximum concentration of substrate consumption (g/L) | Maximum rate of substrate consumption (g/L/h) | Lag time: λ (h) |
| 0.5 | 7,879[a] | 291[a] | 5.8 | 240[a] | 48[a] | 1.2[a] | 0.4 |
| 1.0 | 8,670[a] | 270[a] | 4 | 240[a] | 49[a] | 1.3[a] | 0.5 |
| 2.25 | 6,968[b] | 161[b] | 6.5 | 140[b] | 34[b] | 0.9[b] | 0 |
| 3.0 | 1,790[c] | 59[c] | 15.1 | 40[c] | 16[c] | 0.7[c] | 5.6 |
| 5.0 | 0[d] | 0[d] | 0 | 0[d] | 7[d] | 0.2[d] | 0 |

**Table 2  The kinetics of bacterial growth at different NaCl concentrations.**

| NaCl concentration (%) (w/v) | $X_{max}$ ($\ln(N/N_0)$) | $\mu_{max}$ ($\ln(N/N_0)$/h) | $\lambda$ (h) |
|---|---|---|---|
| 0.5 | 5.200 | 0.771 | 2.037 |
| 1.0 | 5.272 | 0.857 | 2.275 |
| 2.25 | 5.223 | 0.725 | 2.353 |
| 3.0 | 4.561 | 0.569 | 2.658 |
| 5.0 | 3.556 | 0.409 | 2.224 |

stationary phases of bacterial growth and most of the hydrogen was produced during this period.

### Specific H$_2$ production rate and yield

The hydrogen production rate (HPR), $r$ for each NaCl concentration tested was estimated using Eq. (4). The $r$ values of 0.5 and 1.0% NaCl were indistinguishable (Table 1), 2.25 and 3.0% at 140 and 40 mL/h, respectively and 5.0% showed negative value indicating that inhibition had occurred. The logarithm of hydrogen production rate was plotted against the added NaCl concentrations using Eq. (5) and is depicted in Fig. 2. The curve was fitted using a non-competitive inhibition model (Eq. (5)), ($r^2 = 0.99$) at $n = 2$ to demonstrate the relationship of the hydrogen production rate to different NaCl concentrations using a modified Han–Levenspiel model. The model demonstrates that the maximum hydrogen production rate was 374 mL H$_2$/h. Figure 2 shows that H$_2$ production by AM2 is promoted with the addition of 5 to 10.2 g/L (= 0.5 to 1.02% (w/v)) NaCl to the fermentation media. H$_2$ production is inhibited under higher NaCl concentrations and is critically reduced, $C_{crit}$ at 46 g/L (=4.6% (w/v)) NaCl. Media containing 22.5 and 30 g/L NaCl resulted in 15 and 78% reduction of the original amount of H$_2$ produced in 5 g/L NaCl containing media, respectively. The molar yields of hydrogen produced at the end of fermentation (72 h) were compared and presented as mol H$_2$ per mol mannitol. Statistical analysis by Duncan's multiple range test revealed that the H$_2$ yield by cells grown in 5, 10 and 22.5 g/L NaCl supplemented media were not significantly different to each other. The yields were greatly reduced as the NaCl concentration increases.

### Substrate consumption and effects of other fermentative products under different salt concentrations

To determine the effects of various salt levels on substrate consumption, substrate conversion efficiency (SCE) were estimated by dividing the amount of mannitol consumed at a particular time by the amount of initial mannitol (expressed as a percentage). Figure 3 shows the effects of NaCl on mannitol consumption based on the kinetic simulation which was calculated following the method described in *Mu, Yu & Wang (2007)*. At 72 h fermentation, almost all mannitol (95.6 and 98.4%) was consumed in the 0.5 and 1.0% (w/v) NaCl media. Consumption was reduced as the NaCl concentration increases with the highest NaCl concentration (5.0% (w/v)) showing a fluctuating pattern in its substrate consumption.

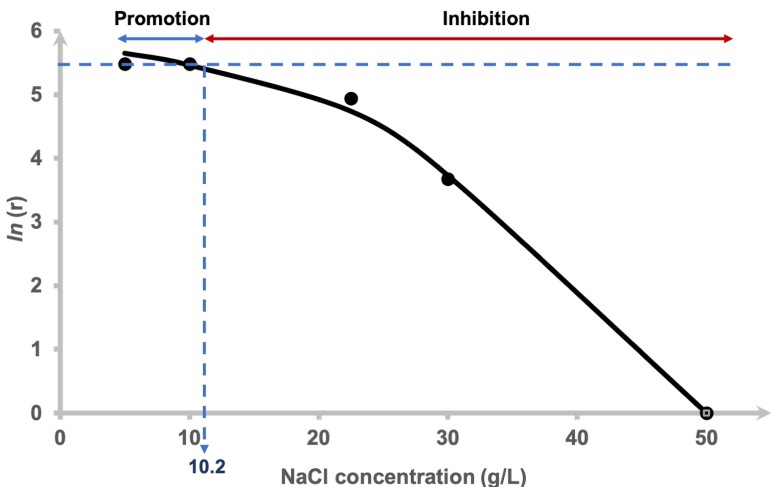

**Figure 2  Effects of NaCl on the hydrogen production rate of *Vibrio tritonius*.** Logarithm of hydrogen production rate, [ln(r)] and relative activity using a non-competitive inhibition model. The fitted curve was drawn in full lines.

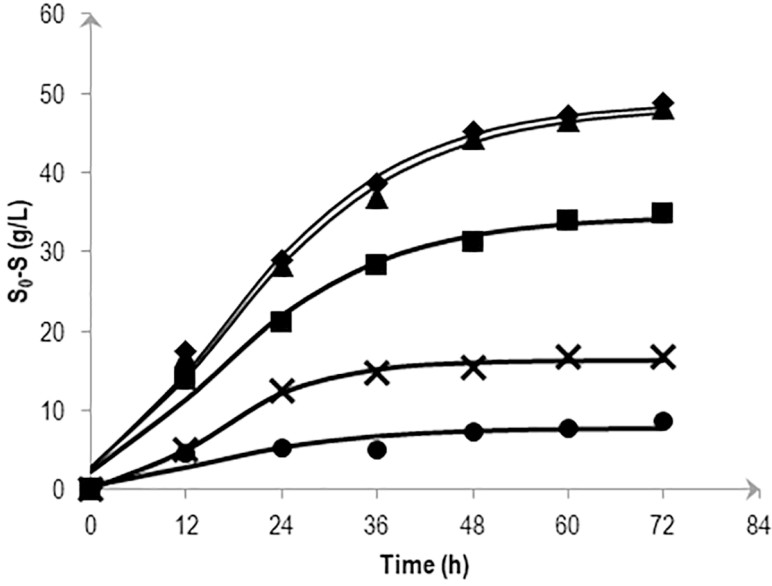

**Figure 3  Effects of NaCl on the mannitol consumption by *Vibrio tritonius* AM2 based on the kinetic simulation.** The media contained 0.5% (♦); 1.0% (▲); 2.25% (■); 3.0% (X) and 5.0% (●) (w/v) of NaCl. Symbols represent the experimental values and the curves were drawn in full lines for all salt levels.

Figure 4 depicts the molar formate concentration throughout the fermentation. For all salt levels tested, formate was accumulated in the first 24 h of fermentation and was later consumed until the end of experiment. Formate accumulation in 5.0% NaCl was significantly delayed and most probably had caused low hydrogen production observed previously. The amount of formate in the first 24 h of fermentation did not show any

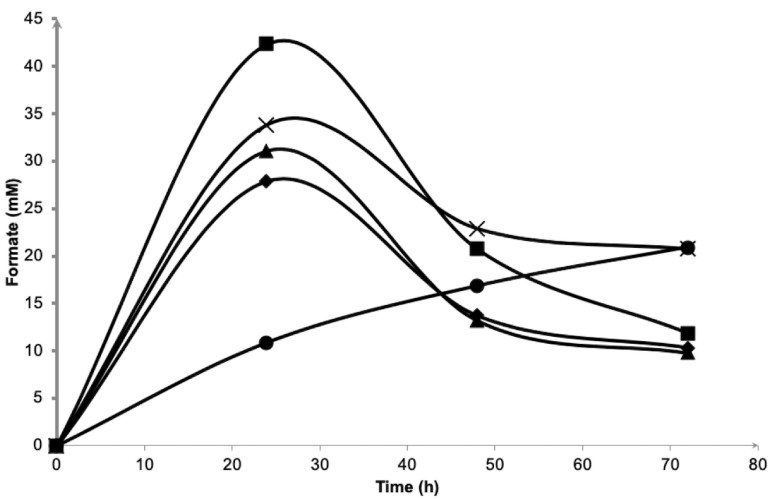

**Figure 4** **Effects of NaCl on extracellular formate emission and re-importation of *Vibrio tritonius*.** The fermentation media contained 0.5% (♦); 1.0% (▲); 2.25% (■); 3.0% (X) and 5.0% (●) (w/v) of NaCl. Symbols represent the experimental values and the curves were drawn in full lines for all salt levels.

**Table 3** **The profile of fermentation products by AM2 at various salt concentrations.**

| NaCl conc. (%) | Molar yields (mmol/mol mannitol) | | | |
|---|---|---|---|---|
| | Acetate | Lactate | Succinate | Ethanol |
| 0.5 | 263 ± 26 | 55 ± 0 | 95 ± 16[a] | 1,610 ± 178 |
| 1.0 | 280 ± 6 | 60 ± 9 | 101 ± 13[a] | 1,469 ± 240 |
| 2.25 | 395 ± 30 | 79 ± 13 | 104 ± 20[a] | 1,664 ± 211 |
| 3.0 | 400 ± 121 | 65 ± 25 | 69 ± 21[b] | 1,517 ± 237 |
| 5.0 | 341 ± 112 | 66 ± 30 | 57 ± 16[b] | 1,330 ± 146 |

differences ($P > 0.05$) among 0.5 to 3.0% NaCl tested with the highest, 2.25% NaCl had achieved 42.3 mM formate. At the end, the average of 10.7 mM formate was left unconsumed/imported in 0.5 to 2.25% NaCl containing media. For both 3.0 and 5.0% NaCl containing media, 20.8 mM formate remained.

Table 3 depicts the molar yields of fermentative products by AM2 in media containing 0.5 to 5.0% NaCl. The ANOVA of raw data shows acetate, lactate and ethanol production was insignificant over the range of NaCl concentrations tested. In relation to cell growth and hydrogen yield, the molar yields of succinate produced were also not significantly different from one another in 0.5, 1.0 and 2.25% NaCl containing media; however, succinate production was significantly reduced up to 3.0% NaCl.

## Fermentative hydrogen production behavior of *Escherichia coli* strain JCM 1649 affected by NaCl and nutrition level of media

The inverted V-shaped pattern in formate production was observed in *E. coli* strain JCM 1649. Maximum formate production was observed after 6 h-cultivation in normal LB broth which contains 0.5% NaCl and LB broth supplemented with 60% artificial water
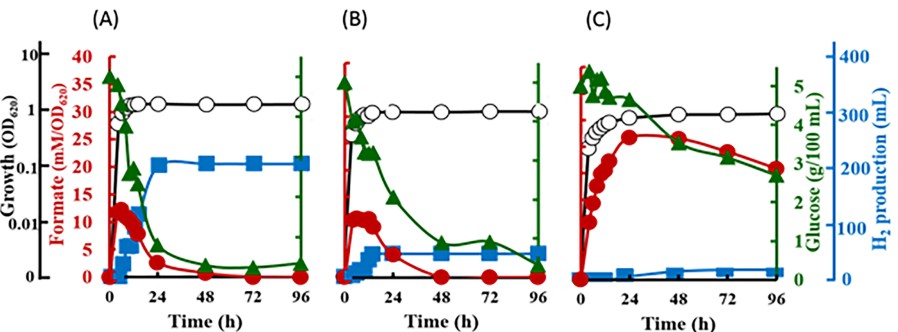

**Figure 5  Time course of formate production, glucose consumption and hydrogen production of *Escherichia coli* strain JCM 1649.** Culture condition was set at pH 6.0 and 37 °C with (A) normal LB broth (with readily available 0.5% (w/v) NaCl); (B) LB broth with 60% ASW (contain 2.3% NaCl); (C) Marine broth (contain 2.25% NaCl). Symbols: ●, formate production; ■, $H_2$ production; glucose consumption; ○, growth.

corresponding to 2.3% NaCl. The formate produced was recorded at 12.2 mM/$OD_{620}$ in 0.5% NaCl LB broth and 11.0 mM/$OD_{620}$ in LB broth with 60% ASW (Figs. 5A–5B). In both cultures, most formate in the supernatant then decreased until 72 h fermentation at which time all formate became undetectable. On the other hand, although the maximum formate concentration in the culture using marine broth reached 26.7 mM/$OD_{620}$ after 24 h-cultivation, the formate were barely consumed by the cells resulting in much lower hydrogen production than those in 0.5% NaCl-LB and LB broth with 60% ASW (Fig. 5C). Note that, hydrogen produced from 0.5% NaCl-LB broth, LB broth with 60% ASW and marine broth were 2,056, 422 and 91 mL/L reactor volume, respectively (Figs. 5A–5C). The rate of sugar consumption after 96 h-cultivation in the 0.5% NaCl-LB broth, LB broth with 60% ASW, and in the marine broth were 93.2%, 93.6% and 46.0%, respectively. Growth was maintained at 1.5, 1.2 and 0.9 $OD_{620}$ after 12, 14 and 48 h-cultivation in the LB broth, LB broth with 60% ASW, and in the marine broth.

## DISCUSSION

Salts are essential in sustaining life on Earth, but higher salt concentrations cause various negative effects in inhibiting major biochemical machineries. The use of seaweed and other marine carbohydrate as alternative fuel sources to terrestrial biomass has proven to be advantageous in biohydrogen production. However, most terrestrial microbial biocatalysts are likely to be less effective for $H_2$ production in conditions with high salt levels that are usually observed when using seaweed biomasses. In fact, the effects of salt on $H_2$ production were confirmed in *E. coli*, where low $H_2$ production was recorded in the use of marine broth, which is a typical medium for marine heterotrophic bacteria (Fig. 5C). Using glucose as substrate, the fermentative hydrogen production by the obligate anaerobe *Clostridium acetobutylicum* was also reduced by the addition of salt at increasing concentrations of NaCl, and 18% reduction under 5 g/L NaCl (*Alshiyab et al., 2008*). Therefore, the cells producing hydrogen have to be halotolerant or capable of adapting to salinity in order to

produce hydrogen when marine biomass is used as feedstock (*Pierra et al., 2014*). As an initial step towards better use of seaweed and other marine carbohydrate as alternative fuel sources, we studied the effect of increasing NaCl concentrations on fermentative hydrogen using a marine vibrio isolated from the sea hare *A. kurodai*.

*V. tritonius* shows great potential in H$_2$ production achieving 80% yields from seaweed powder even in 75% seawater-based medium (*Matsumura et al., 2015*). The bacterium produces up to 1.7 mol H$_2$/mol mannitol and the molar yield of mannitol-grown cells is always higher than those cultured using glucose as a substrate (*Matsumura et al., 2014*). Among the nine gas-producing *Vibrio* spp., hydrogen production efficiency was higher in both species of Porteresiae clade i.e., *V. tritonius* and *V. porteresiae*, and it was comparable to that of enterobacterial species, such as *E. coli*, *Salmonella*, *Enterobacter*, and *Klebsiella* (*Sawabe et al., 2013*). The production kinetics, however, related to salt concentration has never been characterized. Hydrogen production involves complex biochemical reactions via a combination of various fermentation mechanisms (*Wang & Wan, 2009*); thus, the use of kinetic models is the best for describing the relationships between the variables and to explain the behavior of the process quantitatively (*Mu, Wang & Yu, 2006*; *Mullai, Rene & Sridevi, 2013*). The present kinetic reports elucidate the behavior of fermentative hydrogen production by *V. tritonius* AM2 $^T$ at increasing NaCl concentrations in seawater-based media. The bacterium is capable of producing hydrogen under a wide range of salt concentrations from 5 g/L NaCl to 46 g/L NaCl using mannitol as substrate.

Vibrios are phylogenetically related to *Enterobacteriaceae* (*Gomez-Gil et al., 2014*; *Ruimy et al., 1994*), and previously, we showed the structure similarity between vibrio FHL complex and *hyf* genes of *E. coli* (*Matsumura et al., 2014*; *Matsumura et al., 2015*). It is interesting to compare their metabolic pathways and their relationship to bioH$_2$ production. Direct comparison of both bacteria involving cell culture and fermentation, however, requires a few precautions due to the fact that *E. coli* is not a halophile, the requirement of different culture conditions and the bacterium cannot ferment mannitol. In this study, separate investigation on formate-hydrogen conversion by *E. coli* in three different media (normal LB broth, LB broth in 60% ASW and marine broth) helped us to see that hydrogen production by *V. tritonius* was much more stable than that produced from glucose by the *E. coli* strain JCM 1649 in the seawater-based media. *V. tritonius* offers high hydrogen production in saline environments and prolongs stable fermentation in a wide range of NaCl concentrations up to 46 g/L NaCl.

Na $^+$ is one of the important factors in bacterial growth (van Niel et al., 2003; (*Alshiyab et al., 2008*), substrate intake and the yield of fermentative products (*Wei, Tanner & Malaney, 1982*; *Casey et al., 2013*). Generally, growth of Gram-negative non-halophilic bacteria is inhibited under high saline conditions (>12 g/L or >1.2% (w/v) NaCl) due to osmotic stress (*Nakamura, 1977*), which finally affects their fermentative metabolism. As these organisms can experience both osmotic stress and ion toxicity when exposed to high salt concentrations (*Garcia et al., 1997*), the efficiency of fermentative bioH$_2$ is changes depending on the presence of active cells (*Mu, Wang & Yu, 2006*). When compared to the LB supplemented with 0.5% NaCl (Fig. 5A), it is worth to note that only hydrogen production of *E. coli* was inhibited strongly under the presence of 2.30% NaCl (Fig. 5B),

while the formate metabolism, cell growth and glucose consumption are apparently very similar in both media. Later, the fermentation of glucose by *E. coli* in marine broth further shows the influence of other components in the marine-based media to slightly slower cell growth, slower formate metabolism, lower glucose intake and apparent decreases in hydrogen production (Fig. 5C). *Casey et al. (2013)* showed that all six pairs of anions (chloride and sulphate) and cations (sodium, potassium and ammonium) in salts resulted in reduced ethanol fermentation by *Saccharomyces cerevisiae*. Similarly, our study on *E. coli,* using seawater-based media adjusting at various salt levels most likely suggested the effects of not only NaCl but other components in ASW that may contribute to the decrease in hydrogen production. In contrast to these terrestrial microbes, our kinetics studies revealed high and indiscriminate growth patterns of *V. tritonius* in seawater-based medium containing 5 to 22.5 g/L NaCl with elevated and optimum hydrogen production occurring in 5 to 10.0 g/L NaCl (Fig. 2). Though the efficiency of the fermentative system was later reduced at higher NaCl levels, hydrogen production was continuously observed up to the addition of 46 g/L NaCl (Fig. 2). Similar decreases in hydrogen production were also observed in other studies on mixed anaerobic cultures from sewage sludge, *C. acetobutylicum* and *C. butyricum* when increasing concentrations of NaCl in the range of 0 to 30 g/L NaCl are introduced (*Zheng, Zheng & Yu, 2005*; *Alshiyab et al., 2008*; *Lee et al., 2012*). In comparison with *V. tritonius*, hydrogen levels produced in those studies (*Zheng & Yu, 2005*; *Alshiyab et al., 2008*; *Lee et al., 2012*) were much lower and the reducing patterns occurred as early as the addition of 1 g/L NaCl. It is also worth noting that NaCl or salts in general not only originated from the biomass itself but also from the chemicals added during the fermentation (*Casey et al., 2013*; *Palmqvist & Hahn-Hägerdal, 2000*). In our study, the potential use of *V. tritonius* as a biocatalyst for $H_2$ production in high saline environments seems far more suitable in regard to its higher $H_2$ productivity, stability and ability to tolerate a broad-range of salt concentrations.

The inverted V-shaped pattern of formate production in the fermentative systems of both *V. tritonius* (Fig. 4) and *E. coli* JCM 1649 (Fig. 5) indicates the activities of formate being exported out of and then re-imported back to the cells. Formate is not accumulated in the cells but is directly oxidized or exported out of the cells to prevent acidification of the cytoplasm. The transport was via a gating system named FocA (*Leonhartsberger, Korsa & Bo, 2002*) which has been suggested as operating bi-directionally (*Falke et al., 2010*; *Sawers, Blokesch & Böck, 2004*) and is dependent on pH changes and active formate hydrogen lyase (FHL) complex (*Doberenz et al., 2014*; *Lü et al., 2011*). Formate is converted to $H_2$ and $CO_2$ by the FHL complex in the absence of $O_2$ by its substrate (formate) and increased acidity (*Clark, 1989*). A complete genome survey suggested that the strain AM2 has a single FocA homologue protein that most likely assists the re-import of extracellular formate for hydrogen formation (*Matsumura et al., 2015*). In our current experiments, increasing NaCl levels does not seem to affect formate production in *V. tritonius* over the range of 5 to 30 g/L NaCl in seawater-based media (Fig. 4) but the formate re-uptake and hydrogen produced were greatly reduced in media containing ≥22.5 g/L NaCl. Similarly, the formate measured in the *E. coli* fermentative system grown in normal LB broth (5 g/L NaCl) and LB broth prepared with 60% ASW (23 g/L NaCl) were not significantly different but the

hydrogen produced showed significant decreases in the latter medium. Furthermore, the formate production of *E. coli* in marine broth occurs at much slower rates than those in LB broths but after 24 h fermentation, more formate was detected and the amount was much higher than in those produced in the two LB broths (Fig. 5C). Formate re-import process by *E. coli* in marine broth was later suspended (Fig. 5C) resulting in only 9.1 mL hydrogen being produced. Lower nutrition levels under saline conditions further affected the reduction of $H_2$ production of *E. coli*. Our results suggest the effects of not only NaCl but also nutritional levels controlling the FHL activity through the formate re-import providing the substrate of $H_2$.

Up to this point in the study, the observations suggest; (1) the strain AM2 is capable of producing hydrogen under a wide range of NaCl levels, i.e., 5 to 46 g/L NaCl in the seawater-based broth; (2) the amount of formate exported out of the AM2 cells within the first 24 h of incubation is not affected by increasing salt levels up to 30 g/L NaCl but the formate re-import drops significantly at $\geq$22.5 g/L NaCl; (3) the reduced ability of *E. coli* JCM 1649 to convert formate to hydrogen in LB broth prepared with 60% ASW and almost complete shutdown that in marine broth implies the influence of other ionic compounds or components in seawater on the FHL complex or formate intake process; and (4) *V. tritonius* AM2 is a better candidate for fermentative bioH2 production in saline conditions than *E. coli*. Hypothetically, the more substrate consumed (converted to formate) the more hydrogen will be produced. The actual mechanism by which the hydrogen production machineries of vibrios adapt to salt is not yet understood. Careful examination, however, on the *V. tritonius* system revealed that the pattern of hydrogen production (Fig. 1) closely resembles mannitol consumption (Fig. 3). The observation supports our initial hypothesis on substrate-hydrogen conversion and proposes that salt stress is heavily affecting the substrate intake by cells. We later investigated the effects of increasing NaCl concentrations in marine-basal media on substrate consumption by *V. tritonius* AM2. The results support the hypothesis that the highest substrate conversion efficiency (SCE) of 94% and 97% were observed in media containing 5 to 10 g/L NaCl, respectively, which both resulted in the highest hydrogen production rate (HPR) of 240 mL/h hydrogen. The results were in agreement with *Jayasinghearachchi et al. (2010)*, who concluded that addition of 10 g/L NaCl in media resulted in the optimum hydrogen production by *Clostridium amygdalinum* strain C9 in their study.

## CONCLUSIONS

Current research is underway to understand the NaCl-dependent behavior of fermentative hydrogen production of *V. tritonius* strain AM2 via FHL- complex. In conclusion, our data suggests that higher NaCl levels have an overall inhibitory effect on the fermentation system yet *V. tritonius* is still capable of producing hydrogen under a wide range of NaCl concentrations up to 46 g/L NaCl. The fermentative system of *V. tritonius* is generally not affected in seawater-based media containing 5 to 22.5 g/L NaCl, an amount that is relatively higher than common systems of other bacteria. The tolerance to a wider range of NaCl concentrations is a promising attribute towards a better fermentative system

when seawater-based feedstocks or waste biomasses with high salt concentrations are used. This behaviour could also be beneficial in developing continuous batch culture for commercial-scale production. In addition to that, data presented herein (i.e., the kinetics of substrate consumption, formate re-uptake data and succinate production) demonstrates the need for global transcriptome studies that would provide insights in the interaction of cellular reducing power or energy level on hydrogen production. As a result, our data in general suggests the use of bacteria of marine origin, *V. tritonius*, for marine-based fermentative hydrogen production due to its attributes being superior to those of other bacteria.

### Funding
This work was supported by the Strategic Japanese-Brazilian Cooperative Program, Biomass and Bioenergy (Tomoo Sawabe), the JSPS-CAPES bilateral cooperative program (Tomoo Sawabe), and Kaken (26660168 and 16H04976) (Tomoo Sawabe). Nurhidayu Al-saari was supported for her PhD by an education loan from the Malaysia Council of Trust for the Bumiputera (MARA). The funders had no role in study design, data collection and analysis, decision to publish, or preparation of the manuscript.

### Grant Disclosures
The following grant information was disclosed by the authors:
Strategic Japanese-Brazilian Cooperative Program.
Biomass and Bioenergy (Tomoo Sawabe).
JSPS-CAPES bilateral cooperative program (Tomoo Sawabe).
Kaken:  26660168, 16H04976.
Malaysia Council of Trust for the Bumiputera.

### Competing Interests
The authors declare there are no competing interests.

### Author Contributions
- Nurhidayu Al-saari and Eri Amada conceived and designed the experiments, performed the experiments, analyzed the data, prepared figures and/or tables, authored or reviewed drafts of the paper, approved the final draft.
- Yuta Matsumura contributed reagents/materials/analysis tools, approved the final draft.
- Mami Tanaka artwork.
- Sayaka Mino contributed reagents/materials/analysis tools, prepared figures and/or tables.
- Tomoo Sawabe conceived and designed the experiments, analyzed the data, contributed reagents/materials/analysis tools, prepared figures and/or tables, authored or reviewed drafts of the paper.

## Data Availability

Vibrio tritonius strain AM2 is deposited in culture collections at the Japanese Collection of Microorganism (JCM) under JCM16456T and LMG 25401T.

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
