# Peer review of "Understanding the NaCl-dependent behavior of hydrogen production of a marine bacterium, Vibrio tritonius"

_PeerJ, doi:10.7717/peerj.6769_

## Round 0.1 · original submission · Major Revisions

Both reviewers view your manuscript favorably. Please address their observations.

Reviewer 1 ·

Basic reporting

The paper is interesting and acceptable for publication but requires minor revisions

Experimental design

Well done

Validity of the findings

Very good

Additional comments

Minor Revisions
1- In line 137 Materials and Methods, The authors should mention the method they used to install anaerobic condtions in the batch fermentation for hydrogen production by V. tritonius strain AM2 and also for E. coli strain JCM 1649. they should outline the method they used whether by sparging with an inert gas such as nitrogen or argon. If they did not do sparging then they should illustrate this in the paper as both Vibrio and E. coli are facultative anaerobes and can do self instaltion of anaerobiosis in closed system of fermentation. The authors should illustrate which one of these protocols was used.
2- It would be good if the authors mentioned in the abstract and the text the maximum mole of hydrogen produced per mole of mannitol and also compare their results to previous studies results for same and some other carbon sources using same bacterium and some other phylogenetically close bacteria
3- In legend of Figure 5 (line 5) the authors refer to triangles as mannitol consumption while in the Figure per se it is glucose consumption. They should fix this mistake in the legend.
4- In legend of Figure 3 line 1 : put a space between the two words mannitolconsumption>>>>>> mannitol consumption
5- In legend of Figure 5 line 1 : put a space between the two words glucoseconsumption >>>>>> glucose consumption
6- Lines 432 and 440 the expression (terrestrial bacteria) is better to be (other bacteria)
7- Line 317 the expression (terrestrial microbial biocatalysts) should be (microbial biocatalysts other than marine originated Vibrio tritonius).

Reviewer 2 ·

Basic reporting

The use of English should be rechecked. The section of Introduction and Discussion could be revised to make them more concise and easier to comprehend. The comments below might be useful in revising the manuscript.

1) Please check spelling, e.g. favourable (Line 33), controliling (Line 393).

2) Language should be checked throughout the manuscript to improve readability. Also, the use of punctuation marks, e.g. commas, need to be checked.

3) Please revise “Thus, potentially … terrestrial bacteria” (Line 81-84) to improve readability.

4) Line 85-89, and Line 98-102 seem to be irrelevant to the research question. Please consider revising.

5) In Table 1 and 2, if a, b, c, and d were used to indicate significant differences, please place them where appropriate.

6) Please check Line 274-276 and Fig. 3.

Experimental design

The research question was well defined. However, more information and revision are required in order for the manuscript to be suitable for publication. The following comments might be useful in revising the manuscript.

1) Please clarify why E. coli was used as a reference in the present study.

2) Conditions for organic acids and mannitol determination, e.g. mobile phase and flow rate, should be provided (Line 175-178).

3) Eq. (4) was not used in the present study. Please consider revising.

4) Please give details about how microbial growth kinetics, i.e. maximum number of organism, maximum specific growth rate, and lag time, were determined. It would be better if the authors show these results in the manuscript.

5) To my understanding, Eq. 6 is not the modified Monod model reported in the study of Han and Levenspiel (1988). It, instead, shows how the reaction rate constant changes as a function of inhibitor concentration. Please consider revising.

6) Please explain how relative activity (%) in Fig. 2 was calculated.

7) Please explain how substrate conversion efficiency (SCE) was calculated.

Validity of the findings

More explanation and discussion are required for sections of Results and Discussion. However, these sections should be kept concise. The following comments might be addressed to improve the manuscript.

1) Please explain why molar yields of acetate, lactate, succinate, and ethanol were important, so that these should be determined.

2) Although the concentration of formate was determined during the course of fermentation, this might not be adequate to draw a conclusion that hydrogen production by V. tritonius was by FHL-complex. Please explain more on this point.

3) Since E. coli is not a halophile and the conditions for E. coli fermentation were different from those used for V. tritonius, it might be inappropriate to compare the hydrogen production between the two microbial strains (Line 336-338).

4) Please explain more on "Concurrent investigation ... from glucose" (Line 338-340).

5) Please explain more on "Moreover, Casey et al. (2013) ... Saccharomyces cerevisiae" (Line 363-365).

6) Please add a reference for "A genome analysis ... hydrogen formation" (Line 379-381).

7) Conclusions should be rewritten based on the results obtained in the study. Please also consider revising "On the other hand, .... hydrogen production" (Line 435-438) as this was not supported by the results.

Additional comments

The idea of using a marine bacterium to produce hydrogen from marine biomass is interesting as this could alleviate the effect of salt on the bacterium performance. However, the manuscript needs to be revised in order for it to be suitable for publication. More specific comments were given and these might be addressed for the improvement of the manuscript.

---

## Round 0.2 · Minor Revisions

There are still a few minor issues to correct.

Reviewer 1 ·

Basic reporting

Well done

Experimental design

Well done

Validity of the findings

very good

Additional comments

Well done revision

Reviewer 2 ·

Basic reporting

Please recheck carefully the writing to correct typing and grammatical errors.

Experimental design

No comment.

Validity of the findings

Although the authors have made revisions according to the previous comments, I would like the authors to address the following comments for further revisions of the manuscript.

1. Line 227, should Eq. 3 be Eq. 2?

2. Line 243, it would be better if the authors report the growth data of AM2 in the manuscript. This is to show how AM2 responded to the increased NaCl concentration.

3. Line 266, please show the non-competitive inhibition model used to fit the data shown in Fig. 2. Also, please explain why n=2 was used.

4. Line 281-282, the authors stated that substrate conversion efficiency (SCE) was used to determine the effects of various salts level on substrate consumption. However, no SCE (%) is presented in the manuscript. Please consider revising.

5. As far as I am concerned, Fig. 2 is misleading. Since the relative activity was calculated by dividing ln(r) of each NaCl concentration by ln(r) of 0.5% NaCl, the relative activity of 0.5% NaCl should be 100%. I would like to suggest the authors adjust properly the scale of the secondary Y-axis. Please also explain what the gray line in Fig. 2 represents.

6. Please check the results shown in Fig. 4. The authors stated that 2.25% NaCl had achieved 42.3 mM formate (Line 374); however, the data point for 2.25% NaCl at 24 h is at around 37.5 mM.

Additional comments

The manuscript would be suitable for publication after minor revision.

---

## Round 0.3 · accepted · Accept

I am very satisfied with your revision. There is a very minor issue which I have corresponded with the authors about and which I believe can be straightened up directly with our production staff:

"- In line 216, you define relative activity as the ratio of ln (r) at a given concentration and the ln (r) value at 0.5%. I would instead have defined relative activities as r(at a given concentration) divided by r(at 0.5%). Can you please explain your reasoning? It appears to me that it would be clearer if that definition (and its representation as an auxiliary axis in fig. 2) were removed."

Authors have agreed to remove that sui generis definition and provided by email an updated version of their fig.2.

#